# Regulation of Adipose Progenitor Cell Expansion in a Novel Micro-Physiological Model of Human Adipose Tissue Mimicking Fibrotic and Pro-Inflammatory Microenvironments

**DOI:** 10.3390/cells11182798

**Published:** 2022-09-07

**Authors:** Vincent Dani, Solène Bruni-Favier, Bérengère Chignon-Sicard, Agnès Loubat, Alain Doglio, Christian Dani

**Affiliations:** 1Faculté de Médecine, Université Côte d’Azur, INSERM, CNRS, iBV, 06103 Nice, France; 2Clinique Saint Georges, 06105 Nice, France; 3Faculté de Chirurgie Dentaire, Université Côte d’Azur, UPR7354 MICORALIS, UFR Odontologie, 06103 Nice, France; 4Unité de Thérapie Cellulaire et Génique, Centre Hospitalier Universitaire de Nice, 06101 Nice, France

**Keywords:** cell and tissue engineering, adipose tissue, adipose cell progenitors, obesity, inflammation, fibrosis, microphysiological models

## Abstract

The expansion of adipose progenitor cells (APCs) plays an important role in the regeneration of the adipose tissue in physiological and pathological situations. The major role of CD26-expressing APCs in the generation of adipocytes has recently been highlighted, revealing that the CD26 APC subtype displays features of multipotent stem cells, giving rise to CD54- and CD142-expressing preadipocytes. However, a relevant human in vitro model to explore the regulation of the APC subpopulation expansion in lean and obese adipose tissue microenvironments is still lacking. In this work, we describe a novel adipose tissue model, named ExAdEx, that can be obtained from cosmetic surgery wastes. ExAdEx products are adipose tissue units maintaining the characteristics and organization of adipose tissue as it presents in vivo. The model was viable and metabolically active for up to two months and could adopt a pathological-like phenotype. The results revealed that inflammatory and fibrotic microenvironments differentially regulated the expansion of the CD26 APC subpopulation and its CD54 and CD142 APC progenies. The approach used significantly improves the method of generating adipose tissue models, and ExAdEx constitutes a relevant model that could be used to identify pathways promoting the expansion of APCs in physiological and pathological microenvironments.

## 1. Introduction

Lipids begin to accumulate in hepatocytes and muscles when the adipose tissue expansion limit is reached. This ectopic lipid accumulation in non-adipocyte cells causes lipotoxic damage and contributes to insulin resistance, non-alcoholic fatty liver disease and other disorders associated with obesity. Many factors have been reported to be involved in the regulation of adipose tissue expansion, including the maintenance of a pool of adipose progenitor cells (APCs) that are critical to generate new adipocytes in response to physiological and pathological conditions [1]. APCs comprise a heterogeneous group of cell subpopulations that have different degrees of commitment to adipocytes. The developmental hierarchy of APCs described in murine models revealed the major role of CD26-expressing APCs in the generation of adipocytes. The CD26 APC subtype displays features of multipotent stem cells, giving rise to CD54-, identified as ICAM1, and CD142-expressing preadipocytes. Interestingly, CD26, CD54 and CD142 APCs are localized in distinct niches. CD26 APCs are localized in the fluid-filled network surrounding the adipose tissue, i.e., the reticular interstitium, whereas CD54 and CD142 preadipocytes are localized between mature adipocytes [2]. CD26, identified as dipeptidyl peptidase 4 (DPP4), is a ubiquitous glycoprotein bound to the cell membrane that can be cleaved in a soluble form with peptidase activity. CD26 also displays non-enzymatic functions. CD26 is highly expressed in APCs, in which it promotes proliferation by affecting growth factor signaling through an independent peptidase activity [3]. CD26 APCs have a high proliferative capacity and are relatively resistant to differentiation into adipocytes, allowing the renewal of the pool of APCs. The depletion of CD26 APCs in visceral adipose tissues compared to subcutaneous fat in obese mice strongly suggests a link between the low abundance of CD26 APCs and the pathological remodeling of adipose tissue [2]. Therefore, understanding how to promote the expansion of APCs in an obese microenvironment could help to protect against metabolic diseases. However, a relevant in vitro model for studying the expansion of CD26 APCs in physiological and pathological contexts is still missing. 

Different 3D cellular constructs have been developed recently to mimic in vitro the structure and function of tissues. Spheroids are generated from primary cells or immortalized cell lines and are generally composed of only one cell type. Organoids better represent the *in vivo* cellular heterogeneity and physiological functionality of the organ. Organoids originate from tissue-derived adult stem cells, embryonic stem cells or induced pluripotent stem cells, capable of self-renewal and multi/pluripotency. In suitable conditions, these immature cells generate the different phenotypes that are present in the tissue of origin. Finally, according to National Institute of Health (NIH) standards, microphysiological models are 3D constructs composed of multiple human cells, viable in culture for at least 8 weeks, that recapitulate the 3D architecture and cellular composition of the tissue as present *in vivo* [4]. Microphysiological systems (MPS), frequently referred to as organs-on-chips, are interconnected sets of 3D cellular constructs. 

Current 3D adipose-tissue-like models are generated by engineering strategies, basically by assembling APCs and synthetic or natural scaffolds. Vascularized spheroids and organoids have been generated from the stromal vascularized fraction of human adipose tissue [5,6], and decellularized lung matrix has been used to engineer a 3D vascularized adipose tissue construct [7]. Fat-on-chip models have been developed recently (for review see [8]), and different biomaterials have been used to design models of adipose tissue mimicking fibrosis [9,10].

The main limitations of these different adipose tissue substitutes engineered in vitro are as follows: (i) APCs are typically purified from enzymatically dissociated tissues and then expanded in adherent 2D cell culture conditions that rapidly change the natural ratio of APC sub-populations [11], and, (ii) due to the enzymatic dissociation of the tissue, current adipose tissue models lose the native vascularization, the extracellular matrix and critical cell types, such as mature adipocytes. The non-enzymatic dissociation of lipoaspirates, such as emulsification, have been developed to generate micro-fragments of adipose tissues, also named nanofat or microfat [12]. These processes allowing the maintenance of the 3D structure of the tissue are used for reparative and regenerative surgery when a small amount of tissue is required [13]. However, no viable adipocytes were observed in the nanofat samples [12]. The use of some biomaterials allows long-term in vitro culture [14], but biomaterials partially reconstitute the 3D structure and the extracellular matrix, which are known to govern APC behavior and their response to external stimuli [15]. Explants of adipose tissue fragments address the limitations mentioned above and could be the gold standard of the in vitro adipose tissue model. However, the use of explants is seriously limited by 1 to 3 weeks of cell viability in culture [16,17].

We propose a novel adipose tissue model, named ExAdEx (for *Ex vivo*
Adipocyte Expansion) generated by the emulsification of lipoaspirates. The ExAdEx model fulfils the standards of NIH microphyiological model definition, in which APC subpopulations could expand in the cellular complexity and the 3D vascularized extracellular matrix of native adipose tissue. Then, we modified the ExAdEx model derived from healthy adipose tissue to generate a pathological-like model in order to investigate the expansion of APC subpopulations in a fibrotic or inflamed microenvironment.

## 2. Materials and Methods

### 2.1. Human Samples

The method was developed from lipoaspirates of subcutaneous adipose tissues obtained from elective cosmetic surgery on 27 healthy women donors aged from 21 to 75 years old, with a mean age of 40 years and a body mass index of 23.4 kg/m^2^. Donors had not reported any disease. 

### 2.2. Isolation of the Infranatant Cell Pellet Depleted from Red Blood Cells

Lipoaspirated adipose tissues were processed as previously described [12]. Basically, lipoaspirates were centrifuged at 3000 rpm for 3 min, resulting in separation into four layers: a superior layer containing oil, a middle layer consisting of compacted yellow adipose tissue, the infranatant fluid composed of infiltration liquids and a pellet containing APCs and red blood cells (RBCs). Infranatant cells were treated with 10 volumes of ammonium chloride (NH4Cl)-containing buffer (RBC Lysing buffer, Invitrogen, Villebon-sur-Yvette, France) for 5 min at 37 °C. Then, the effect of the RBC lysis buffer was stopped by adding 20 volumes of PBS, and the cells were centrifuged at 3000 rpm for 3 min. RBC and APC viabilities were measured with the Count and Viability Assay Kit of the Muse^®^ Cell Analyzer according to the manufacturer’s instructions (Luminex Corporation, Austin, TX, USA).

### 2.3. Emulsification of the Adipose Tissue Fraction

The adipose tissue fraction was mechanically emulsified by shifting it between two 10 cc syringes connected to each other by a Luer Lock connector. After 30 passages, the emulsified tissue was mixed with the infranatant cells prepared as described above and maintained in EGM-plus media (CC-4133 from Promocell, Heidelberg, Germany) for the indicated times.

### 2.4. In Vitro Maintenance of the ExAdEx Model, Beiging and Treatment with TNFα or TGFβ1

ExAdEx was maintained in Ultra Low Attachment (ULA) flasks or plates (Corning, Avon, France) in EGM-plus media at 37 °C/5% CO_2_ on a shaken balance. APC expansion was visualized in situ by EdU (PK-CA724-594FM from Promocell, Heidelberg, Germany). Briefly, ExAdEx tissues were maintained ex vivo for 14 days and then incubated overnight with EdU 10 μM. Samples were fixed, and EdU incorporation into proliferating cells was detected using a Click-iT EdU Alexa Fluo reaction and imaging kit according to the manufacturer’s instructions (Molecular Probes™ #C10339). Images were acquired with an LSM 710 confocal microscope in z-stack mode and reconstructed by ImageJ v1.52p (Bethesda, MD, USA). For beiging, 100 mg of the model expanded for 4 weeks was encapsulated in agarose as previously described [18] and maintained for 2 weeks in EGM-plus medium supplemented with 1 µM rosiglitazone, 2 µM T3 and 2.5 µg/mL insulin. To generate inflamed and fibrotic ExAdEx models, the healthy ExadEx model expanded for 4 weeks was maintained for 7 days in EBM (Cat: C-22211; PromoCell, Heidelberg, Germany) supplemented with 2% FCS, 22.5 μg/mL ascorbic acid, 1 μg/mL heparin and 10 ng/mL TNFα (Cat: H8916, Sigma-Aldrich, St. Louis, MO, USA) or 5 ng/mL TGFβ1 (Cat: 100-21, Preprotech, Neuilly-sur-Seine, France).

### 2.5. Viability and Functional Assays

#### 2.5.1. Lactate Dehydrogenase Activity (LDH)

The cell viability of the model was measured using the lactate dehydrogenase release assay. One hundred milligrams of ExAdEx-processed tissue were cultured in a well of a ULA 24-well plate, and the culture supernatant was collected 24 h after a culture medium change. Tissues were then treated with 10% Triton X-100 for 4 h to obtain the maximum level of LDH release and to calculate the percentage of cytotoxicity for each assay. LDH release assays were conducted according to the manufacturer’s instructions (LDH-Glo Cytotoxicity Assay, Promega, #J2380).

#### 2.5.2. Measurement of Adiponectin and IL-6 Secretion

Levels of secreted adiponectin and interleukin 6 (IL-6) were measured using ELISA. Conditioned media were collected 24 h after the medium was changed and stored at −80 °C. Adiponectin and IL-6 doses were determined using commercial ELISA kits (R & D Systems, Minneapolis, MN, USA, #DY1065 and #DY206, respectively) according to the manufacturer’s instructions.

#### 2.5.3. Lipolysis

Tissues were washed once with PBS and maintained for 24 h in RPMI supplemented with 2% fatty acid free BSA, 5 µM triascin C and 1 ng/mL ascorbic acid. Lipolysis was stimulated with 1 µM isoproterenol for 2 h or 24 h. Glycerol release into the culture medium was determined as an index of lipolysis according to the manufacturer’s instructions using a Glycerol Detection Assay (Promega, Madison, WI, USA, #J3150).

### 2.6. RNA Extraction and Reverse Transcription Quantitative Polymerase Chain Reaction

One hundred milligrams of tissue was disrupted using TissueLyser LT, and total RNA was extracted using the RNeasy Plus Universal kit (Qiagen, Hilden, Germany) according to the manufacturer’s instructions. Reverse transcription–polymerase chain reaction (RT–PCR) analysis and real-time PCR assays were conducted as previously described [19]. All primer sequences are detailed in Appendix A. The expression level of each gene target relative to control was calculated using the 2^−ΔΔCt^ method. The housekeeping gene *TATA box-binding protein (TBP)* was used as internal control to normalize individual sample differences as previously described [19]. 

### 2.7. Isolation of APCs and Flux Cytometry Analysis

Collagenase digestion was only used to quantify APC expansion or to investigate the impact of inflammation and fibrotic environment on APC subtypes. In brief, 100 mg of ExAdEx model was dissociated for 45 min at 37 °C in 2 mL PBS containing 2 mg/mL collagenase A (Sigma 10103586001) and 20 mg/mL bovine serum-albumin fatty acid free (Sigma A6003). Then, APCs were separated from the tissue by low-speed centrifugation (200× *g*, 10 min), resuspended in EGM and seeded as a 2D monolayer culture. Adherent cells were identified as APCs based on their immunophenotype (positive for CD105, CD90, CD73) and their capacity to differentiate into adipocytes (see Appendix A). APCs were counted or fixed with 4% formaldehyde for 15 min the day after plating for FACS analysis with CD26 (Clone BA5b/FITC; Ozyme, Saint-Cyr-l’École, France), CD54 (Clone 1H4/APC; molecular probe) and CD142 (Clone HTF-1/PE; Invitrogen, Villebon Sur Yvette, France).

### 2.8. Confocal Microscopy and Second-Harmonic Generation (SHG) Imaging

ExAdEx samples were fixed with 4% PAF and then incubated with primary anti-collagen 1 (Abcam ab260043), anti–elastin (Abcam ab21610), anti-laminin (ab11575), anti-fibronectin Santa Cruz sc8422) and anti-CD31 (Abcam ab28364) antibodies overnight at 4 °C and then with the corresponding secondary antibody for 45 min at room temperature. Lipid droplets were stained with Oil Red O (ORO), and nuclei were stained with DAPI. Samples were visualized on an LSM 780 NLO inverted Axio Observer Z1 confocal microscope (Carl Zeiss Microscopy GmbH, Jena, Germany) using a Plan Apo 25× multi immersion (oil, glycerol, water) NA 0.8 objective. The SHG light source was a Mai Tai DeepSee (Newport Corp., Irvine, CA, USA) tuned to 880 nm. A forward SHG signal was detected with an oil condenser (1.4 NA), bandpass filter 440/40 nm and transmission PMT. Backward SHG was collected with a GaAsP (BIG) nondescanned module of 440/10 nm.

### 2.9. Statistical Analysis

The results are presented as the mean ± SEM. To determine statistical significance, the results were analyzed using GraphPad Prism version 9. Groups were compared using the Wilcoxon signed-rank test with *n* < 10 per group or using the two-tailed unpaired Student’s *t*-test to compare group means, with *n* > 10 per group, unless indicated otherwise. All data are shown, and no outlier removal was performed. For all data, statistical significance was defined as * *p* ≤ 0.05; ** *p* ≤ 0.01; *** *p* ≤ 0.001.

## 3. Results and Discussion

### 3.1. Generation of the ExAdEx Model Derived from the Adipose Tissue of Healthy Donors

A schema of the method we applied to generate ExAdEx is shown in Figure 1a. First, we hypothesized that the movement of the cannula by surgeons during liposuction helps to mechanically dislodge CD26 cells from the reticular interstitial niche and therefore release them in the infranatant, the fluid fraction of the lipoaspirate discarded by practitioners because it contains blood cells. However, some authors previously claimed that a portion of adipose-derived cells are released in the infranatant [20]. Indeed, we show that CD26 APCs were dramatically enriched in the infranatant fraction compared to the tissue fraction. In contrast, CD54 APCs were primarily localized in the tissue fraction (Figure 1b). The differential distribution of CD26 and CD54 APCs in the infranatant and tissue fractions, respectively, fits well with their anatomical localization as mentioned above [2]. The innovativeness of the process to generate the model also stems from the use of emulsified tissues as a natural bioactive matrix for CD26 APC expansion. The rationale for non-enzymatic dissociation, such as the emulsification step, is, first, that it conserves the native 3D structure of the tissue [21] and, second, that previous publications have reported the ability of mechanical stimuli to activate quiescent progenitor cells to proliferate, to restore tissue homeostasis [22]. In agreement with this claim, no proliferative cells were detectable in the lipoaspirate, whereas a high number of Edu-positive cells were detectable in the tissue after emulsification (Figure 1c). The final step of the process was then to combine cells from the infranatant fraction with the emulsified tissue fraction to generate the ExAdEx product. To quantify the expansion of APCs, the collagenase digestion of the ExAdEx products was performed in experimental conditions previously described to isolate APCs from adipose tissue [23]. Adherent cells derived from ExAdEx upon collagenase digestion were identified as APCs, based on their immunophenotype and their capacity to differentiate into adipocytes (see Appendix A). As shown in Figure 1d, the total APC population could be expanded by more than 18-fold after 2 weeks in in vitro culture, as shown by producing ExAdEx from 27 donors aged from 21 to 75 years old, with a mean age of 40 years and a BMI of 23.4 kg/m^2^. At that stage, the abundance of CD26 APCs in the total APC population was 36%, whereas the percentage of CD54 APCs was 4%.

Due to the emulsification method allowing minimal changes of the microscopic structure of the lipoaspirate tissue [21], APC expansion occurred in a microenvironment wherein the extracellular matrix and the endothelial cell networks of the native adipose tissue were conserved (Figure 2).

The viability and functional activities of the model were then investigated. An analysis of LDH released in the culture medium, reflecting dead or damaged cells [24], showed that less than 20% of the maximum LDH release was detectable after 60 days in culture, whereas 90 days appeared to be the limit of the tissue viability, as two ExAdEx products on three tested presented more than 20% of the maximum release (Figure 3a). These data support the long-term viability of the model over a period of 8 weeks. This characteristic contrasts with the viability of explants of adipose tissue that can be maintained *ex vivo* for only 14 days [16]. Moreover, ExAdEx remained metabolically functional during the long-term period in culture, as revealed by the levels of secretion of the anti-diabetic adipokine adiponectin [25], which were similar in the lipoaspirate and the derived ExAdEx products, and the lipolysis capacity measured by glycerol release upon β-adrenergic receptor stimulation [26] (Figure 3b,c). Finally, when maintained in appropriate culture conditions, ExAdEx conserved the physiological potential of beiging, i.e., the capacity to generate clusters of brown-like adipocytes dispersed in white adipose tissue [27] (Figure 3d). Interestingly, adipocytes with lipid droplet diameters of approximately 100 μm could be observed in the ExAdEx model maintained for 14 days to 60 days in vitro (see Figure 1c, Figure 2 and Figure 3c,d). This lipid droplet size is similar to the size that can be observed in native tissue and is substantially larger than the 20–40 μm lipid droplets generated in other 3D models of adipocyte spheroids [6] or by micropatterning preadipocytes [28]. Altogether, the data support the claim that the ExAdEx model significantly improves on the existing models of human adipose tissue.

### 3.2. Regulation of APC Expansion in the Pathological-like ExAdEx Models 

In order to investigate the regulation of APC subpopulations in pathological microenvironments, fibrosis and inflammation, both hallmarks of obesity [29], were induced separately from the ExAdEx model. Inflammation and fibrosis were modeled by treatment of ExAdEx with tumor necrosis factor-alpha (TNFα) and transforming growth factor beta 1 (TGFβ1), respectively. The rationale for the choice of these cytokines is that the elevated secretion of TNFα by macrophages has been shown to play a critical role in the low-grade chronic inflammation of obese adipose tissue [30]. We and others have previously shown that TGFβ superfamily members, such as TGFβ1, secreted by macrophages are key mediators that promote the fibrosis of adipose tissue [31]. The TNFα treatment promoted major changes in the ExAdEx products, resembling those previously reported for obese adipose tissue: adipocyte gene expression, such as *PLIN1* and *GLUT4*, was reduced, as was the secretion of adiponectin (Figure 4a,b). The proinflammatory cytokine IL6 was secreted at low levels in the untreated condition, suggesting a weak inflammatory state in the healthy ExAdEx model. In contrast, IL6 secretion was dramatically increased in the inflammatory model and interestingly could be reversed in the presence of the anti-inflammatory molecule dexamethasone (Figure 4b). The remodeling of the extracellular matrix in the inflamed ExAdEx model was also visualized by laminin staining (Figure 4c). The treatment of ExAdEx with TGFβ1 to generate a fibrotic-like model induced the inhibition of *PLIN1* and *GLUT4* gene expression, the secretion of adiponectin (Figure 5a), and the stimulation of *INHBA* gene expression, a marker of APCs having the potential to differentiate into myofibroblasts, a source of fibrosis [31] (Figure 4a). TGFβ1, but not TNFα, treatment induced *MMP14* gene expression (Figure 5b), which has been previously reported to be increased in the early stages of obesity [32]. Finally, TGFβ1 treatment promoted the remodeling of the extracellular matrix, as revealed by the appearance of fibrotic collagen fibers observed with second-harmonic generation imaging (Figure 5c). All of these data support an inflammatory- and fibrotic-like microenvironment in the pathological ExAdEx model.

The impact of the pathological-like microenvironment on the percentages of CD26, CD54 and CD142 APC subpopulations was then addressed by flow cytometry. The gating strategy used to quantify the different APC subpopulations is shown in Appendix A. The data showed that the abundance of APC subpopulations was differentially regulated in the fibrotic and inflamed ExAdEx models (Figure 6). The abundance of the CD26^+^/CD54^−^/CD142^−^ subpopulation, denoted CD26^+^, was inhibited in both inflammatory and fibrotic-like microenvironments. It is interesting to note that TGFβ1 has been shown to promote the proliferation of CD26 cells when cells were purified from adipose tissue [2], whereas CD26 cell abundance was dramatically reduced in the TGFβ1-treated ExAdEx model. This difference observed for the TGFβ1 effects underlines the relevance of analyzing APC regulation in the fully adipose tissue microenvironment. The CD26+/CD142+ population was also inhibited in both pathological models, whereas the CD26+/CD54+ population was increased in the inflamed microenvironment but not in the fibrotic microenvironment. In contrast, the abundance of the CD54^+^ subpopulation increased in both conditions. The CD142^+^ subpopulation was reduced by inflammation but increased by fibrosis, whereas the CD54+/CD142+ population increased in the inflamed microenvironment only. A schema reporting the effects of TNFα and TGFβ1 treatments on APC subpopulations is presented in Appendix A. Recently, Raajendiran et al. reported APC subpopulations giving rise to adipocytes with divergent metabolic capacities [33]. The impact on the proliferative and adipogenic capacities of the CD54 and CD142 subpopulations, as well as on the metabolic and endocrine capacities of their adipocyte progenies, remains to be investigated. However, knowing the potential of APCs to undergo differentiation into myofibroblasts when exposed to an inflammatory or fibrotic environment, it is tempting to speculate that the CD54 subpopulation will display a profibrogenic/proinflammatory phenotype in the pathological model. The accumulation of CD54^+^ APCs in the subcutaneous adipose tissue of high-fat-diet-induced obese mice without a corresponding increase in mature adipocyte differentiation supports this hypothesis [34]. CD142, also named tissue factor, plays a critical role in the activation of the coagulation cascade that facilitates blood clot formation. Its overexpression has been reported in obese adipose tissue, and its role has been suggested in developing inflammatory obesity (for review see [35]). The role of CD142 APCs in adipogenesis is more controversial because this subpopulation is described as fully adipogenic or as an inhibitor of adipogenic differentiation via paracrine effects [36].

## 4. Limitations of the Model and of the Study

One goal of the ExAdEx process was to avoid the enzymatic dissociation of the liposuction samples in order to maintain the 3D structure and the cell complexity of the tissue as pre-sent in vivo. However, it is likely that the liposuction process causes damage to the tissue. The low levels of *INHBA* gene expression and of IL-6 secretion strongly suggest that tissue damage does not promote the myofibroblast differentiation of APCs and the inflammation of the model. However, it would be interesting to perform the ExAdEx process from surgical biopsies to avoid liposuction damage to the tissue and to improve the physiological relevance of our model. The ExAdEx model derived from the liposuction of subcutaneous adipose tissue from healthy donors was then modified to generate a pathological-like model. The latter model was not generated from the adipose tissue of obese patients for two main reasons: first, adipose tissue samples can easily be collected from healthy donors undergoing elective cosmetic liposuction surgery, and because the samples are plastic surgery waste, donors are only asked to sign a routine consent form. Liposuctions are not performed on obese patients. A biopsy of visceral adipose tissue is a clinical procedure with much more complex regulatory requirements, precluding the simple and regular sourcing of obese adipose tissue samples. In addition, fibrosis and inflammation, hallmarks of obesity, can be induced separately and in a controlled manner from healthy samples. Finally, the regulation of APC subpopulations can be investigated and compared in healthy and pathological micro-environments derived from the same donor. However, it would be interesting to generate the ExAdEx-pathological model from obese patients to increase the pathological relevance of our model. Works are in progress in our team to perform the ExAdEx process from the surgical biopsies of subcutaneous and visceral adipose tissues from obese patients.

## 5. Conclusions

ExAdEx thus represents a robust model of human adipose tissue with which fully integrated responses could be investigated. The model reveals that the fibrotic and inflamed microenvironments have different impacts on APC subpopulations. It could help in identifying mechanisms that promote the expansion of APCs and a more in-depth functional characterization of APC subpopulations in healthy and obese-like contexts. More generally, ExAdEx is a relevant human adipose tissue model that could be used as an in vitro preclinical platform in the early stages of pharmaceutical drug testing to accelerate the pharmaceutical development of molecules to counteract the pathological remodeling of human adipose tissue. The ability of brown/beige adipose tissues to actively drain circulating glucose and triglycerides and oxidize them could prevent hyperglycemia and hypertriglyceridemia. Therefore, these tissues represent promising targets to counteract obesity and associated metabolic diseases [37]. As shown, we observed that the ExAdEx model has the potential to generate UCP1-expressing adipocytes. The ExAdEx models could be optimized to generate beige adipocytes and could represent a powerful in vitro model to investigate pathways promoting beige adipocytes in human white adipose tissue. In the long term, the ExAdEx process could be the means to generate an abundant source of human beige adipocytes to graft as part of a tissue-based therapy against obesity.

## 6. Patents

Two patents result from the work presented in this manuscript. “Method for the in vitro or ex vivo amplification of human adipose tissue stem cells” (Publication number: 20220098552) and “Method for the in vitro or ex vivo amplification of stem cells of brown or beige adipocytes” (WO2022/013404 A1).

## Figures and Tables

**Figure 1 cells-11-02798-f001:**
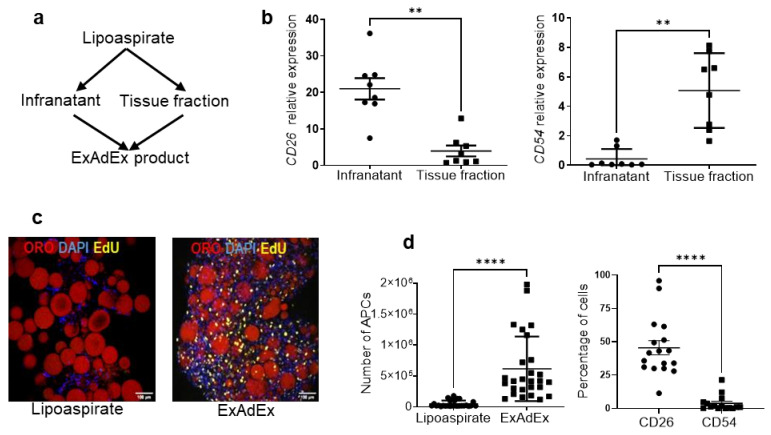
Generation of the ExAdEx model and its main characteristics. (**a**) A schema of the ExAdEx process. Cells were isolated from the infranatant of the lipoaspirates and then mixed with the tissue fraction previously dissociated mechanically. (**b**) CD26 and CD54 gene expression in the infranatant and the tissue fraction; *n* = 8 donors. (**c**) The ExAdEx process promotes cell proliferation. The lipoaspirate or the emulsified product after 14 days in culture was incubated with EdU for 24 h. Then the EdU click-iT-labeling of the EdU incorporated into cells was performed and confocal imaged. Red: ORO for lipid droplets; Yellow: Edu-labeled nuclei. Blue: DAPI for nuclei. Scale bar: 100 μm. (**d**) Expansion of APC subpopulations. The number of total APCs was quantified from 100 mg of lipoaspirate and from the derived ExAdEx model maintained for 14 days in culture; *n* = 27 donors (left panel). Then, the percentage of CD26^+^/CD54^−^ and CD54^+^/CD26^−^ APCs was quantified by FACS analysis; *n* = 15 donors (right panel). ** *p* ≤ 0.01; **** *p* ≤ 0.001.

**Figure 2 cells-11-02798-f002:**
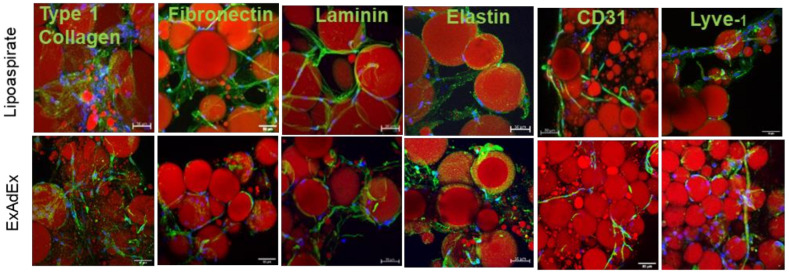
Maintenance of the extracellular matrix and the endothelial cell network. The lipoaspirate and the ExAdEx product were labeled for the indicated proteins of the extracellular matrix. The vessel and lymphatic endothelial cell networks were visualized with CD31 and Lyve-1 labeling, respectively. The lipoaspirate was fixed after 1 day in culture, whereas the ExAdEx product was fixed after 60 days. Scale bar: 50 μm.

**Figure 3 cells-11-02798-f003:**
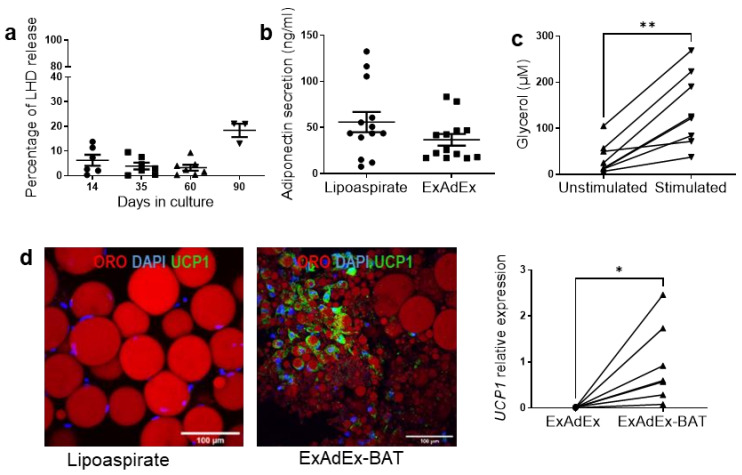
Long-term viability and functionality. (**a**) Levels of LDH activity released in the culture medium were determined after the indicated times in culture. Data are presented as a percentage of the LHD activity of the triton-lysed tissue, measuring the maximum released; *n* = 7 donors. (**b**) Long-term functionality. Levels of adiponectin secreted in 24 h by 100 mg of lipoaspirate and the derived ExAdEx product maintained in vitro for 60 days; *n* = 13 donors. (**c**) Lipo-lytic capacity. One mg of ExAdEx encapsulated was maintained in vitro for 21 days, and then the glycerol release was measured 24 h after stimulation with 1 μM isoproterenol; *n* = 7 donors. Similar data were obtained after maintenance 60 days in culture. (**d**) Beiging potential. The ExAdEx products were maintained in vitro for 30 days and then were induced to undergo beiging as described in the Methods section. After two weeks, UCP1 was revealed by immunofluorescence (left panel, scale bar: 100 μm) and by gene expression compared to the lipoaspirate (right panel). Data are expressed as a fold increase compared to UCP1 expression in lipoaspirates; *n* = 7 donors. * *p* ≤ 0.05, ** *p* ≤ 0.01 using the Mann–Whitney test.

**Figure 4 cells-11-02798-f004:**
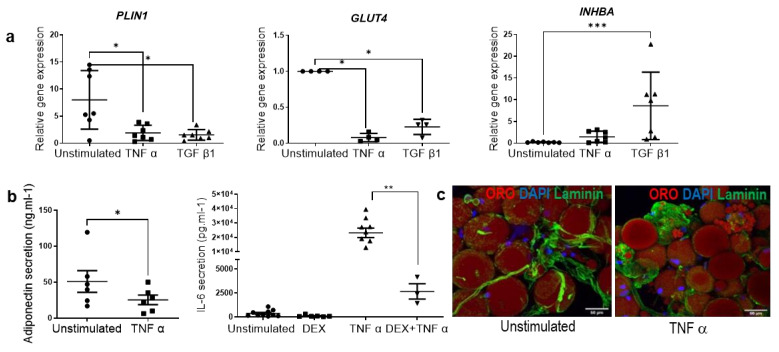
Generation of the ExAdEx inflamed-like model. (**a**) Regulation of *PLIN1*, *GLUT4* and *INHBA* gene expression in ExAdEx–TNFα and ExAdEx–TGFβ1 models. (**b**) Levels of adiponectin secretions. One hundred mg of ExAdEx was maintained in vitro for 30 days and then treated or not with 10 ng/mL TNFα for 7 days. Levels of secretion during the last 24 h of the treatment were determined by ELISA; *n* = 6 donors. Levels of IL6 secretions. One hundred mg of ExAdEx was maintained in vitro for 30 days and then treated or not with 0.25 μM dexamethasone or 10 ng/mL TNFα or both for 7 days. Levels of secretion during the last 24 h of the treatment were determined by ELISA; *n* = 9 donors. * *p* ≤ 0.05; ** *p* ≤ 0.01; *** *p* ≤ 0.001 using the Mann–Whitney test. (**c**) The remodeling of the extracellular matrix in the inflamed ExAdEx model. The ExAdEx and ExAdEx-TNFα models were fixed and stained with anti-laminin antibodies and with ORO for lipid droplets.

**Figure 5 cells-11-02798-f005:**
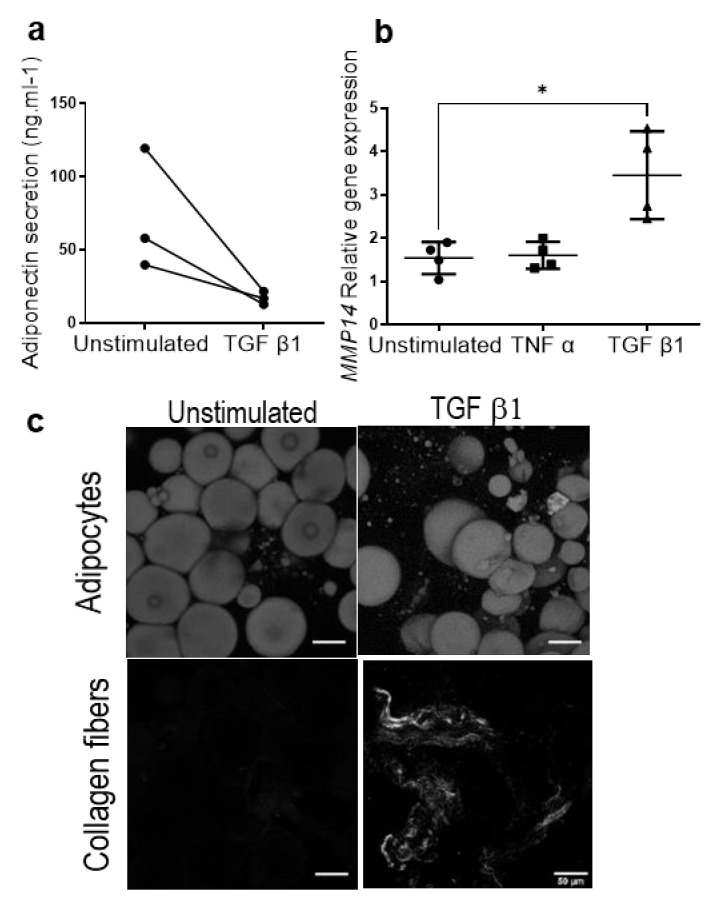
Generation of the ExAdEx fibrotic-like model. (**a**) Levels of secreted adiponectin. Ten mg of the ExAdEx model was treated with 5 ng/mL TGFβ1 for 7 days, and the levels of secretion during the last 24 h of the treatment were determined by ELISA; *n* = 3 donors. (**b**) The expression of the *MMP14* gene determined by qPCR; *n* = 4 donors. * *p* ≤ 0.05. (**c**) The visualization of fibrotic collagen fibers by SHG imaging. Scale bar: 50 nm.

**Figure 6 cells-11-02798-f006:**
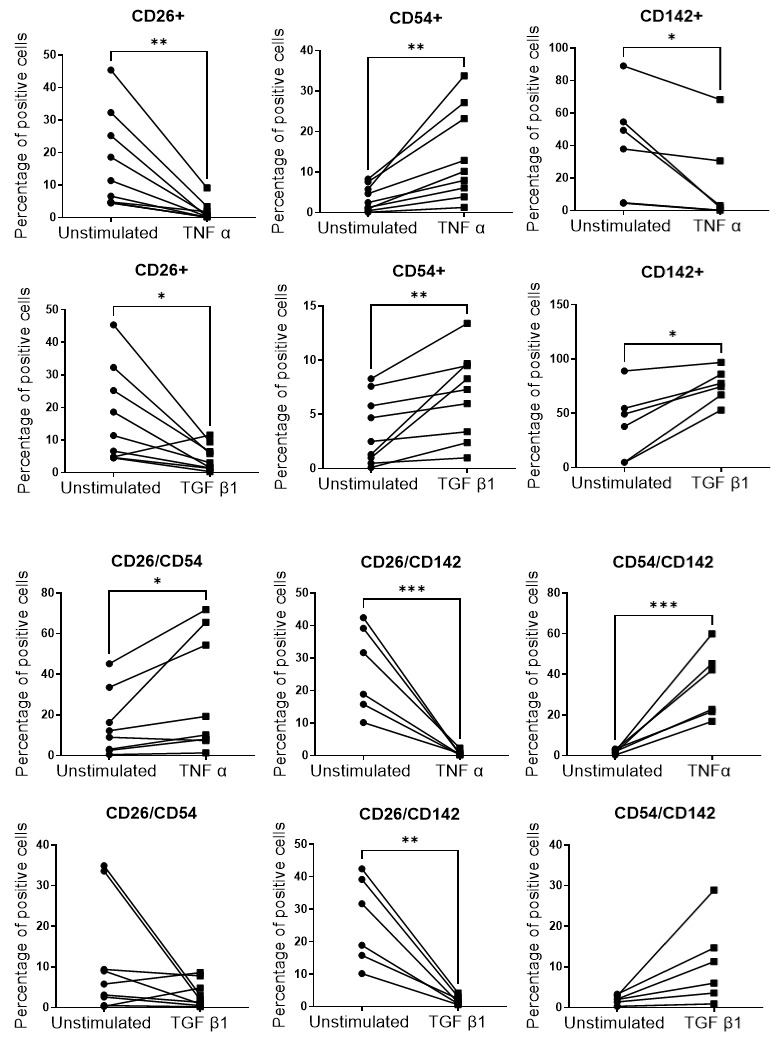
Percentages of APC subtypes in untreated and TNF α or TGF β1-treated Ex-AdEx models; *n* = 5–9 donors. * *p* ≤ 0.05; ** *p* ≤ 0.01; *** *p* ≤ 0.001.

## Data Availability

Not applicable.

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
