# Peer review of "Regulation of Adipose Progenitor Cell Expansion in a Novel Micro-Physiological Model of Human Adipose Tissue Mimicking Fibrotic and Pro-Inflammatory Microenvironments"

_cells, 2022, doi:10.3390/cells11182798_

Round 1

Reviewer 1 Report

Dani et al. describes a novel technique for culturing human subcutaneous adipose tissue samples which were obtained from cosmetic liposuction. The culture technique "ExAdEx" is intended to avoid collagenase dissociation of the extracellular matrix, which the authors suggest changes the natural environment of the adipose progenitor cells too much to provide a valid assay. Instead, the authors developed a technique where adipose progenitor cells and other cells from the donated adipose tissue remained in culture for as long as 60 days, without losing function or viability. The model was also successful to evoke inflammatory and fibrotic phenotypes, which might allow scientists to study various disease processes where adipose tissue may be involved. The authors note that they have two patents based on this novel culture technique.

Major points:

1) Please rework the results section so that the figures are clear and easily read. As they stand in the draft paper, the font is too small to be easily read and the photomicrographs are too small for critical assessment. Figure 1 might either be expanded to being full page-sized or broken into more figures with fewer sections and larger font, photomicrographs and graphs. 

2) Please consider including most of the supplementary figures in the paper; these are mentioned with the primary figures and would enhance the story. To my assessment, only supplementary figure 5 and table 1 are truly supplementary.

3) The paper lacks explicit referencing, in particular in the methods section and in the results/discussion section. Please be more explicit with your references.

4) The discussion provides only a superficial comparison of the new culture model with the range of 2D and 3D alternatives. In this regard, the discussion reads as more of a promotional brochure for the product than as a traditional scientific discussion of the results in light of the current literature.

5) Please provide a critical assessment of the limitations of the study and the new culture model of adipose tissue in the discussion. Between the poor discussion of how the model compares to other techniques and the lack of critical assessment of the model limitations, the paper loses credibility.

6) Following on from the previous 2 suggestions, the paper would benefit from a critical, honest discussion of how damaging liposuction is to adipose tissue. The novel method is promoted for avoiding the trauma of collagenase dissociation of the tissue biopsy, but the nature of liposuction must surely cause significant damage in and of itself. Does liposuction evoke myofibroblast differentiation, for example, as the tissue is wounded and healing? Would this technique be even better with surgical biopsies? For example, samples of visceral fat taken during surgeries. While the authors are not requested to use their method to culture visceral fat from a human or non-human donor, use for this purpose does warrant discussion and would excite more readers. 

7) The conclusion offers the suggestion that the new culture may be used to study human disease processes and as a tool for drug testing. This is helpful but surely the authors could expand this section to offer more ideas of where their new model would benefit the scientific community.

Minor points:

1) The paper has minor editorial errors - inconsistent use of italics for 'in vitro' (etc.), missing punctuation or brackets, abbreviations not explained at their first use. The manuscript is very well written, however, and these issues are not common.

2) Please include missing details in the methods such as which type of collagenase was used, and the source of the enzyme. As written, the methods are clear but cannot be reproduced due to the lack of such details. References to support the various techniques would also be very helpful.

3) Please consider providing more details about the donors... gender, any diseases, etc.

Reviewer 2 Report

General Comments

This submission is well referenced and well written.  The authors provide detailed description of a novel 3D methodology for studies of human adipose tissue ex vivo which provides new insights and research opportunities.  The focus on the CD26 population in the lipoaspirate infranatant is of particular importance, at least from this reviewer’s perspective.   The insights into both the white and beige features of the model are likewise important.  The authors might wish to consider the following general and specific comments:

1.      The Discussion section should incorporate some level of review of the terminology now being used to describe 3D Microphysiological systems as opposed to organoid and/or microfragmented fat models of human adipose biology.  The literature is entering a phase where these terms are not well defined and there is likely to be confusion moving forward in distinguishing among such models.  By highlighting this topic in this important paper, the authors can help set the future context on how these terms should be employed.

2.      The Discussion may be strengthened by indicating future directions by which the 3D model can be employed for research and discovery while also acknowledging any potential limitations its currently faces or might need to address.

Specific Comments

Ln 78.  The authors should mention the Institutional Review Board approved protocol and the regulatory oversight authority that oversaw the bioethics of this project at this juncture.  This information is essential to the project description and publication.

Ln 91.  The authors describe the emulsification of the lipoaspirate tissue.  For many in the literature today, 3D MPS (3 dimensional Microphysiological systems) is the term used to describe cell scaffold complexes reconstituted from enzymatically isolated adipose-derived cells and a biomaterial matrix.  The emulsified tissues are frequently described as either microfragmented adipose tissue or fat or as organoids.  The authors need to acknowledge that these terminologies are in use as they present their work to the reader. The discussion of the terminology should be explicitly stated in either the Introduction or Discussion at the authors’ discretion.  In this context, it would be prudent for the authors to cite relevant prior publications in the literature describing microfragmented and other mechanically disrupted adipose organoid models that have performed 3D modeling.

Ln 136.  Provide a reference to the original primary research article describing the 2 delta delta CT quantification method.

Ln 145, 191.  Change “thanks” to “based on”.

Ln 197. Change “Thanks” to “Due to”.

Ln 207.  There are reports demonstrating that ex vivo lipoaspirate explants can maintain viability for periods well in excess of 2 weeks and such studies should be acknowledged in addition to ref 7.

Ln 309.  It may be helpful to indicate to the reader at this juncture that CD142 is equivalent to “tissue factor” and possibly discuss its implications with respect to clotting mechanisms in the context of healthy vs metabolically unhealthy adipose tissue/obesity and its associated co-morbidities.

Author Response

Please see the attchment

Round 2

Reviewer 1 Report

I would like to thank the authors for their revisions, which in my opinion, make your paper a stronger article which will have greater impact. My compliments especially on the revised figures, which look excellent. The immunoflourescent images are quite beautiful.

Reviewer 2 Report

The authors have provide a thorough response to this reviewer's comments.  The revised version is deemed ready for publication.